# Exploring Canine Mammary Cancer through Liquid Biopsy: Proteomic Profiling of Small Extracellular Vesicles

**DOI:** 10.3390/cancers16142562

**Published:** 2024-07-17

**Authors:** Adriana Alonso Novais, Guilherme Henrique Tamarindo, Luryan Mikaelly Minotti Melo, Beatriz Castilho Balieiro, Daniela Nóbrega, Gislaine dos Santos, Schaienni Fontoura Saldanha, Fabiana Ferreira de Souza, Luiz Gustavo de Almeida Chuffa, Shay Bracha, Debora Aparecida Pires de Campos Zuccari

**Affiliations:** 1Institute of Health Science (ICS), Universidade Federal de Mato Grosso (UFMT), Sinop 78550-728, MT, Brazil; adriana.novais@ufmt.br (A.A.N.); luryanmikaely@hotmail.com (L.M.M.M.); 2Brazilian Biosciences National Laboratory, Brazilian Center for Research in Energy and Materials (CNPEM), Campinas 13083-100, SP, Brazil; guilherme.tamarindo@lnbio.cnpem.br; 3Molecular Investigation of Cancer Laboratory (MICL), Department of Molecular Biology, Faculdade de Medicina de São José do Rio Preto/(FAMERP), São José do Rio Preto 15090-000, SP, Brazil; bia-balieiro@hotmail.com; 4Pat Animal Laboratory, São José do Rio Preto 15070-000, SP, Brazil; danielanobrega@patanimal.com.br; 5Laboratory of Molecular Morphophysiology and Development (LMMD/ZMV), University of São Paulo, Pirassununga 13635-900, SP, Brazil; gislainesantos@usp.br (G.d.S.); schaiennifontouras@usp.br (S.F.S.); 6Department of Veterinary Surgery and Animal Reproduction, School of Veterinary Medicine and Animal Science, FMVZ, São Paulo State University (UNESP), Botucatu 18618-681, SP, Brazil; fabiana.f.souza@unesp.br; 7Department of Structural and Functional Biology, Institute of Biosciences, UNESP—São Paulo State University, Botucatu 18618-689, SP, Brazil; luiz-gustavo.chuffa@unesp.br; 8Department of Veterinary Clinical Sciences, College of Veterinary Medicine, Ohio State University, Columbus, OH 43210, USA; bracha.2@osu.edu

**Keywords:** canine, biomarker, proteomics, small extracellular vesicles, liquid biopsy, mammary carcinoma, diagnosis

## Abstract

**Simple Summary:**

We studied canine mammary tumors to better understand similar human breast cancer using a technique called liquid biopsy, which analyzes blood samples to detect disease, focusing on the detection of tiny particles called small extracellular vesicles. These structures are very interesting because they can carry proteins that may indicate the presence of cancer. In this study, we collected blood from healthy dogs, dogs with benign and malignant CMTs, and those in remission and also with recurrence. We found no differences in the size or amount of the vesicles among the groups but identified specific proteins that could serve as markers for cancer. These proteins could potentially help in the diagnosis, prognosis and monitoring of mammary cancer.

**Abstract:**

(Background). Canine mammary tumors (CMTs) have emerged as an important model for understanding pathophysiological aspects of human disease. Liquid biopsy (LB), which relies on blood-borne biomarkers and offers minimal invasiveness, holds promise for reflecting the disease status of patients. Small extracellular vesicles (SEVs) and their protein cargo have recently gained attention as potential tools for disease screening and monitoring. (Objectives). This study aimed to isolate SEVs from canine patients and analyze their proteomic profile to assess their diagnostic and prognostic potential. (Methods). Plasma samples were collected from female dogs grouped into CMT (malignant and benign), healthy controls, relapse, and remission groups. SEVs were isolated and characterized using ultracentrifugation (UC), nanoparticle tracking analysis (NTA) and transmission electron microscopy (TEM). Proteomic analysis of circulating SEVs was conducted using liquid chromatography–mass spectrometry (LC–MS). (Results). While no significant differences were observed in the concentration and size of exosomes among the studied groups, proteomic profiling revealed important variations. Mass spectrometry identified exclusive proteins that could serve as potential biomarkers for mammary cancer. These included Inter-alpha-trypsin inhibitor heavy chain (ITIH2 and ITI4), phosphopyruvate hydratase or alpha enolase (ENO1), eukaryotic translation elongation factor 2 (eEF2), actin (ACTB), transthyretin (TTR), beta-2-glycoprotein 1 (APOH) and gelsolin (GSN) found in female dogs with malignant tumors. Additionally, vitamin D-binding protein (VDBP), also known as group-specific component (GC), was identified as a protein present during remission. (Conclusions). The results underscore the potential of proteins found in SEVs as valuable biomarkers in CMTs. Despite the lack of differences in vesicle concentration and size between the groups, the analysis of protein content revealed promising markers with potential applications in CMT diagnosis and monitoring. These findings suggest a novel approach in the development of more precise and effective diagnostic tools for this challenging clinical condition.

## 1. Introduction

Canine mammary tumors (CMTs) have garnered significant attention in veterinary medicine, not only due to their high prevalence among female dogs but also because of their intriguing similarities with human breast cancer. As the most common tumor in intact female dogs, CMTs provide a unique opportunity to bridge the gap between veterinary and human medicine, offering insights into shared mechanisms and potential breakthroughs. In addition to anatomical and histological similarities, CMTs and human breast cancer share genetic and molecular features, making the former an invaluable comparative model for understanding the complexities of both diseases. Studying the canine counterpart can provide essential understanding into disease progression, treatment strategies, and the development of potential biomarkers that are applicable to both species [1,2]. 

In both veterinary and human medicine, the identification of reliable biomarkers plays a crucial role in the early detection, accurate diagnosis, and effective prognosis of disease. Biomarkers serve as indicative tools for the presence or progression of diseases [3,4], making their exploration in CMTs not only beneficial for our beloved canine companions but also significant for advancing our understanding of human breast cancer. 

Tissue biopsy limitations are increasingly recognized in precision medicine. In contrast, liquid biopsy (LB) offers minimal invasiveness, easy sample acquisition, and dynamic analysis, enabling the early detection of malignant tumors [5]. LB relies on unique biomarkers in circulation, which reflect the patient’s disease status [6]. 

One promising avenue in biomarker discovery involves investigating the cargo of small extracellular vesicles (SEVs). These small membrane-bound structures released by cells carry a payload of proteins, nucleic acids, and other bioactive molecules [6,7]. Exploring the extracellular vesicle cargo in CMTs may unveil specific proteins usable as biomarkers, revolutionizing diagnosis and prognosis [8,9,10], as well as distinct molecular pathways deregulated in the disease that could be further investigated as a therapeutic strategy. By embracing the convergence of veterinary and human medicine, we propose a scientific endeavor that not only benefits our companion animals but also advances cancer research, with far-reaching implications for both species. In this study, we performed an exploratory analysis and evaluated, using mass spectrometry, the proteomic profile of SEVs in dogs bearing mammary tumors, either benign or malignant, before and after mastectomy, under relapse or remission.

## 2. Materials and Methods

### 2.1. Experimental Groups

Forty adult female dogs were distributed into the following groups—(G1) eight female dogs with malignant mammary neoplasia (simple carcinomas); (G2) eight female dogs with benign mammary neoplasia (carcinoma in mixed tumor); (G3) eight female dogs with tumor relapse after mastectomy (metastasis); and (G4) eight female dogs in clinical remission (over a year following mastectomy of malignant tumors); and (G5) eight healthy female controls. For Groups G1 and G2, blood was collected at the time of diagnosis of mammary tumor (G1.1 and G2.1) and at the time of surgical suture removal (G1.2 and G2.2), approximately 10 days after mastectomy. Groups G3, G4, and G5 had only one collection. The collection of Group G3 (healthy control individuals) was performed during annual routine consultations; the collection of Group G4 individuals was performed at the time of confirmation of recurrence (tumor metastasis), during the veterinary consultation, and the collection of Group G5 patients was performed at least 1 year after mastectomy, during the patient’s return to the veterinary office for medical reevaluation, characterizing remission of the disease. A summary of our methodological approach is shown in Figure 1. The average age of the female dogs in all groups was 8.5 years, with mixed-breed dogs being the most prevalent among all the breeds. 

The inclusion criteria for the neoplastic group were adult dogs with a clinical and histological diagnosis of CMTs, with or without metastasis. The control group consisted of healthy adult spayed patients without comorbidities or treatments. Histopathological grading was performed using the Goldschmidt classification (2011) [11], which is based on Nottingham’s classification.

### 2.2. Liquid Biopsy

Three to five milliliters of blood was collected into tubes with EDTA anticoagulant and immediately centrifuged twice at 2500× *g* at 23 °C for 15 min to obtain platelet-poor plasma. Subsequently, the plasma underwent three consecutive centrifugations at 4 °C, first at 300× *g* for 10 min to remove cells, then at 2500× *g* for 10 min to remove cellular debris, and finally at 16,500× *g* for 30 min to remove small extracellular vesicles (SEVs) larger than 150 nm. After five centrifugation cycles, the plasma was aliquoted and stored at −80 °C for the subsequent isolation and characterization of the SEVs.

### 2.3. Tissue Samples (Tumor Fragment) and Histopathology

The tumor fragment obtained during surgical excision was fixed in 10% formaldehyde for 24 h and then transferred to 70% alcohol for subsequent histopathological diagnosis. Histopathological analysis of the canine patients was conducted at the PatAnimal Laboratory in São José do Rio Preto, SP. The samples were processed following the standard protocol, which included sectioning in a microtome and staining with hematoxylin and eosin (HE). The slides were examined by two independent veterinary pathologists using a common Nikon E20 optical microscope (Nikon Corporation, Tokyo, Japan).

### 2.4. Exosomes Isolation

Exosome isolation was performed using ultracentrifugation (UC) (Optima XE-90 Ultracentrifuge; rotor 70 Ti; Beckman Coulter, Indianapolis, IN 46268 United States) at the Laboratory of Molecular Morphophysiology and Development (LMMD/ZMV), University of São Paulo, Pirassununga, SP, Brazil. To obtain exosome-enriched content, the pellets were filtered through a 0.22 μm pore filter and ultracentrifuged twice at 119,700× *g* for 70 min at 4 °C. Following the second ultracentrifugation, the pellet was diluted in phosphate-buffered saline (PBS) and stored at −80 °C.

### 2.5. Nanoparticle Tracking Analysis (NTA)

The pellets containing SEVs were characterized for size, morphology, and quantity. Initially, they were diluted in 50 µL of magnesium–calcium-free PBS at a dilution factor of 1:500 in PBS. A laser beam was passed through a sampling chamber containing this liquid suspension. The suspended particles scattered the light, allowing for visualization under a microscope at 20× magnification. A video camera (sCMOS in Camera Level 15 at a temperature of 37 °C) captured five images of 30 s of these particles in Brownian motion. Using the Nano-Sight software (NS300; NTA 3.4 Build 3.4.003; Malvern, Worcestershire, UK), the particles were individually tracked, and the hydrodynamic diameter of the particles was calculated using the Stokes–Einstein equation.

### 2.6. Transmission Electron Microscopy (TEM) 

Following ultracentrifugation, the pellet was diluted in a fixative solution (0.1 M sodium cacodylate, 2.5% Glutaraldehyde, 4% paraformaldehyde, pH between 7.2 and 7.4) for 2 h at room temperature (RT). Then, 2 mL of ultrapure water was added followed by centrifugation at 119,700× *g* for 70 min at 4 °C to remove the fixative solution. The pellet was diluted in 20 μL of ultrapure water and stored at 4 °C. The contents were placed on a copper grid coated with pioloform for 5 min, and the excess was removed with moist filter paper. The grid was then inserted into a drop of 2% aqueous uranyl acetate for 3 min, removing the excess again with moist filter paper. Transmission electron microscopy (FEI 200 kV, model Tecnai 20, emitter LAB6) was performed at the Multiuser Electron Microscopy Laboratory, USP, Ribeirão Preto, Brazil. Images were captured at 140,000× magnification.

### 2.7. Proteomics Analysis

Samples containing EVs isolated from plasma were evaluated by mass spectrometry (LC-MS/MS) at the Multiuser Proteomics, Metabolomics and Lipidomics Laboratory at ESALQ/USP, Piracicaba Campus. To this end, the EV pellets isolated from plasma samples needed to go through a pre-processing step, which was carried out at the Animal Reproduction Laboratory at UNESP, Botucatu Campus, SP. Initially, the samples were submitted to sonication. This step included the lysis of EV samples diluted in PBS with RIPA, followed by the sonication procedure for each sample. A final volume of 500 μL was determined and the concentration was standardized to 10 × 10^6^ EVs/sample containing 50 mM TRIS-HCl pH 7.2, RIPA buffer and protease inhibitors (Aprotinin 1 µg/mL, Leupeptin-1 µg/mL, PMSF 35 µg/mL and EDTA 0.8 mM). Samples were sonicated (Sonifier Sound Encloresure Ultrasonics, Shanghai, China) for three cycles of 30 s followed by a pause of 30 s and, after sonication, they were centrifuged at 10,000× *g* for 30 min at 4 °C, and the supernatant was collected. Next, the supernatant was washed and concentrated in ultrafilters (3 kDa cutoff, Amicon Ultra, Millipore Corporation, Merck KGaA, Darmstadt, Germany) at room temperature, using 50 mmol ammonium bicarbonate buffer. The total protein concentration of the samples was determined by Bradford protein assay (Pierce™ 660 nm Protein Assay, ThermoFisher Scientific, Pierce Biotechnology, Rockford, IL, USA) using a NanoDrop 2000 (Fisher Scientific Term™ Wilmington, DE, USA). Following, the samples were submitted to protein digestion and sample preparation (spots/bands) for proteomics. Initially, 4 μg of protein were loaded and the bands were cut into fragments of approximately 1 mm^3^, using a small scalpel blade and a glass Petri dish, previously rinsed with methanol and milliQ water. Gel fragments were washed until the complete removal of the dye and then dehydrated with 100% acetonitrile (ACN) (200 μL/spot) for 10 min. The gel fragments were rehydrated by adding reducing solution (20 mM dithiothreitol—DTT, BioRad, cat# 161-0611, MW = 154.3 = [30 mg/10 mL of 50 mM AmBic solution = [0.2 g/50 mL MS-grade H_2_O] (40 μL/spot)), and the samples were kept at 56 °C for 40 min. Next, an alkylation solution (55 mM iodoacetamide—IAA, GE, cat# RPN 6302V, MW = 185.0 = 102 mg/10 mL of 50 mM AmBic solution) was added to cover the gel fragments (40 μL/spot), and the samples were incubated in the dark at room temperature for 30 min. For trypsin digestion, 15 μL of trypsin solution (Promega Trypsin, Part No. V511) (20 ng/μL) was added onto the dehydrated gel fragments. The samples were left for 15 min at 4 °C so that the trypsin penetrated the fragments. Excess trypsin was removed and 50 mM of AmBic was added until the spots were completely covered (40 μL), and the samples were incubated at 37 °C for 14 h. Finally, samples were submitted to the elution process. The final solution containing the peptides eluted in gel was concentrated in a vacuum concentrator, and samples were resuspended in 10 μL TFA 0.1% (v:v) in MS-grade H_2_O (Sigm-Aldrich, cat# 299537, CAS# 76-05-1), vortexed and desalted with the aid of the Zip Tip, before being eluted in 10 μL of 50% (v:v) ACN in MS-grade H20 and transferred to vials.

LC–MS was performed on an NanoElute (Bruker Daltonik, Bremen, Germany) system coupled online with a hybrid TIMS-quadrupole TOF mass spectrometer [12] (Bruker Daltonik timsTOF Pro, Germany) via a nano-electrospray ion source (Bruker Daltonik Captive Spray). The peptides were resuspended in 10 µL water with 0.1% formic acid and separated on an Aurora column 15 cm × 75 µm ID and an 1.9 µm reversed-phase column (Ion Opticks) at a flow rate of 500 nL min^−1^ in an oven compartment heated to 50 °C. To analyze samples from whole-proteome digests, we used a gradient starting with a linear increase from 2% B to 35% B over 18 min, followed by further linear increase to 95% B in 2 min, which was held constant for 2 min (22 min total run). The column was equilibrated using four volumes of solvent A. The mass spectrometer was operated in data-dependent PASEF [13] mode with one survey TIMS-MS and four PASEF MS/MS scans per acquisition cycle. We analyzed an ion mobility range from 1/K0 = 1.3 to 0.85 vs. cm^−2^ using equal ion accumulations and ramp times in the dual TIMS analyzer of 100 ms each. Suitable precursor ions for MS/MS analysis were isolated in a window of 2 Th for *m*/*z* < 700 and 3 Th for *m*/*z* > 700 by rapidly switching the quadrupole position in sync with the elution of precursors from the TIMS device. The collision energy was lowered stepwise as a function of increasing ion mobility, starting from 27 eV for 1/K0 = 0.85 vs. cm^−2^ and 45 eV for 1/K0 = 1.3 vs. cm^−2^. We made use of the *m*/*z* and ion mobility information to exclude singly charged precursor ions with a polygon filter mask and further used ‘dynamic exclusion’ to avoid the re-sequencing of precursors that reached a ‘target value’ of 20,000 a.u. The ion mobility dimension was calibrated linearly using three ions from the Agilent ESI LC/MS tuning mix (*m*/*z*, 1/K0: 622.0289, 0.9848 vs. cm^−2^; 922.0097, 1.1895 vs. cm^−2^; and 1221.9906, 1.3820 vs. cm^−2^).

The raw files (.d) were processed in MaxQuant software, version 2.4.0.0., which is specific for protein analysis based on mass spectrometry [14]. The search engine integrated into MaxQuant is Andromeda [15]. The default configuration was used for data obtained from mass spectrometers using TimsTof Pro technology. The fragment ion mass tolerance was 0.5 Da. The enzymatic specificity was for trypsin with tolerance for peptides with up to two undigested cleavage sites. The methionine oxidation (15.994915 Da) and N-terminal acetylation of proteins (42.010565 Da) were defined as variable modifications, and cysteine carbamidomethylation (57.021464 Da) was defined as fixed modification. The minimum peptide length was seven amino acids. The false discovery rates (FDRs) of peptides and proteins were 1%. We also considered at least one unique peptide for protein identification as a criterion. At the end of this data processing, we obtained the relative abundance values of all identified proteins. The database used was UP000805418 obtained from https://www.uniprot.org/proteomes?query=canis+lupus+familiarisX (taxonomy *Canis lupus familiaris*). The download date was 9 March 2023, with a total of 43,622 proteins available. Perseus analysis software version 2.0.9.0 [16] was used to filter the proteomic results. Matrix proteins identified only by one modification site, as well as those identified by the reverse database, and possible contaminants were excluded from subsequent analyses. Proteins were filtered so that only those that had values greater than zero in at least 90% of the samples from at least one of the groups remained in the abundance matrix. Afterwards, a script in the R programming language (https://www.R-project.org/) was used to refine the filter, based on the percentage of protein presence in the groups, and normalize the data by the total ion count (TIC).

Processed peak intensities were then further submitted to Metaboanalyst 5.0 [17] to generate heatmaps after Pareto scaling, presenting only the 25 most significant proteins. Confirmed proteins retrieved from the Uniprot database for *Canis lupus familiaris* (CANF) were displayed using their gene names, while predicted, inferred, uncertain or Ig-like proteins were indicated by their primary accession. Exclusive and shared proteins were visualized using Venn diagrams (https://bioinformatics.psb.ugent.be/webtools/Venn/, accessed on 16 January 2024). The PANTHER Overrepresentation Test [13] was performed using the GO database (17 January 2024 release) and *Canis lupus familiaris* gene set as a reference. Only the unique features found in each group were analyzed by the Fisher test, and outputs with a false discovery rate (FDR) < 0.05 were considered enriched. The −log (*p* value) was reported. To assess protein interaction networks, the protein list was submitted to the STRING database [18]. Finally, exclusive proteins were double-checked in the Uniprot database for *Canis lupus familiaris* (CANF), and those that retrieved scores as ‘experimental evidence at protein level’, ‘experimental evidence at transcript level’, ‘protein inferred from homology’ or ‘protein predicted’ were inserted in squares. Uncertain proteins were excluded, as well as Ig-like proteins, since they did not retrieve any specific output.

### 2.8. Statistical Analysis 

For the non-omics data, values were submitted to Shapiro-Wilk and Kolmogorov-Smirnov normality tests. For the 1st and 2nd collection sets, values were parametric and underwent to paired *t*-test. For the other sets, Ordinary one-way ANOVA test was performed followed by Tukey’s multiple comparisons test for parametric data or Kruskal-Wallis for non-parametric. Values were expressed as mean and standard deviation (SD).

## 3. Results

### 3.1. Extracellular Vesicles Isolated from Canine Plasma Exhibited Characteristics Typical of Small Extracellular Vesicles (SEVs), with Consistent Concentration and Size across Different Groups

The average concentration and size of SEVs for each canine group are presented in Figure 2. While there were not statistically differences in SEV concentration and size among most groups, the recurrence and remission groups showed distinct patterns. This suggests parameters may not offer significant diagnostic or prognostic value independently. It highlights the importance of considering a more comprehensive approach or exploring other SEV characteristics, such as cargo content, for a more thorough assessment. SEVs were isolated from plasma using ultracentrifugation following a standard procedure described in previous studies and characterized according to the guidelines of the International Society for Extracellular Vesicles (ISEV) [19]. Validation was conducted using NTA (Figure 2A–H) and TEM (Figure 2I), revealing a typical ‘donut-like’ appearance by TEM. Our data collectively indicate that the SEVs isolated from canine plasma exhibit characteristics typical of exosomes. 

### 3.2. SEVs’ Proteomic Profile from Dogs with Malignant Tumors Exhibit a Larger Number of Unique Proteins, Most of Them Associated with Immune and Wound Healing Processes

We initially investigated whether the proteomic profile of SEVs differed among healthy (G5), benign (G2.1), and malignant (G1.1) conditions. The heatmap (Figure 3A) reveals distinct protein clusters specific to each group, with SEVs from dogs bearing malignant tumors containing a higher number of exclusive proteins or proteins with increased intensity. Indeed, this group showed the highest number of total (85) and unique (31) proteins (Figure 3B), while the benign group had 49 total and 1 unique protein, and the healthy group showed 56 total and 2 unique proteins. The squares in Figure 3B represent only the exclusive proteins. These findings suggest that the cargo composition between the pathological status is sharply distinct, potentially serving as biomarkers for disease staging. Gene ontology enrichment analysis was performed using all outputs from data processing, revealing 14 significantly enriched biological processes in the malignant group, most of which were related to immune or wound healing (Figure 3C). The unique proteins identified appeared to cluster and form a network with interactions, as depicted in Figure 3D.

### 3.3. SEV’s Cargo from Dogs with Malignant Tumor Decreases after Mastectomy, and This Trend Persists Even after Surgery in Cases of Remission or Relapse

Having identified a specific cargo in SEVs, including unique proteins, we interrogated whether mastectomy would alter the proteomic profile in benign or malignant conditions. For this purpose, we performed a paired analysis before and after surgery, as summarized in Figure 4. The heat map shows distinct clusters for benign (Figure 4A) and malignant (Figure 4B) conditions, while unique proteins identified before and after surgery are detailed in Figure 4C,D. In the benign group, 9 proteins were exclusive in SEVs cargo before surgery and 10 were exclusive after. Conversely, in the malignant group, 39 proteins were found in SEVs collected before surgery compared to 3 after, indicating that mastectomy remarkably decreased the number of proteins in SEVs, particularly in the malignant group. Gene ontology analysis retrieved significantly enriched biological processes only in the malignant group, including pathways involved in cholesterol and lipid transport, in addition to immune-related processes (Figure 4E).

Next, we examined whether proteins found in malignant condition would change upon remission or relapse (Figure 5). We observed that SEVs from dogs with malignant tumors had increased peak intensity for most proteins, while these proteins were decreased or absent in the relapse or remission groups, as shown in the heatmap (Figure 5A). This was further confirmed by the increase in the number of exclusive proteins (51) found in the malignant group (Figure 5B) compared to the relapse (3) and remission (1) groups. Once again, most of the enriched biological processes were related to lipid and cholesterol metabolism, innate and adaptative immune pathways, wound healing, coagulation, and response to stress (Figure 5C). The majority of identified proteins formed a network of interactions (Figure 5D). The difference between the relapse and remission groups is described in Figure 6. Both groups showed clustered proteins, but most of them were not validated or were primarily Ig-like proteins (Figure 6A), indicating a possible regulation of the immune system. Both groups exhibited exclusive proteins (nine for relapse and four for remission), but no significant enrichment output was obtained from the gene ontology analysis. 

## 4. Discussion

Our findings demonstrate a diverse array of proteins present in each investigated condition. As depicted in Figure 3, the heatmap reveals distinct patterns of protein expression between healthy dogs and those with benign and malignant tumors, suggesting significant alterations in the proteomic content of SEVs in different pathological states. Additionally, the Venn diagram highlights exclusive proteins found in each group, providing information on potential molecular markers associated with each condition. Gene ontology enrichment analysis reveals specific biological pathways and molecular functions that may be affected in malignant tumors, indicating potential therapeutic targets. Similarly, Figure 4, Figure 5 and Figure 6 provide valuable information on changes in the proteomic profile of SEVs at different stages of the disease and treatment response. These results suggest that SEVs may play a significant role in the progression of canine mammary cancer and could serve as useful biomarkers for monitoring treatment efficacy and predicting patient prognosis. 

EVs have emerged as pivotal mediators of intercellular communication and are increasingly recognized as potential source of biomarkers for liquid biopsy in various cancer types [20]. They exhibit selectivity in tumor cell signaling and are continuously secreted into body fluids from the early stages of the disease [20,21]. Aguilera-Rojas and colleagues (2018) [21] isolated SEVs from the blood serum of dogs and reported a nanoparticle concentration (×E10/mL) ranging from 107.4 +/− 6.8 to 403.2 +/− 25.8 in non-cancer patients and 225.6 +/− 10.4 to 500.4 +/− 76.4 in cancer patients, respectively. However, no significant difference was observed between non-cancerous and cancerous samples, despite notable variation within individual samples [22]. Similarly, our results demonstrated that the SEV concentration did not exhibit a statistically significant difference in the canine cohorts, either before or after the surgical excision of the primary tumor. In fact, the concentration before mastectomy was higher than after mastectomy, considering individual values. However, SEV concentration in the remission, relapse, and control groups was even higher than that obtained for the mammary tumor groups. Like Aguilera-Rojas and colleagues (2018) [21], we also noted considerable variation within individual samples, despite the lack of statistical significance. The constant influx of SEVs, the dynamic release-uptake of different cells, and the lack of a precise characterization of SEV origin make it challenging to determine whether the quantity of SEVs in cancer patients differs significantly from that in healthy control patients. 

On the contrary, scientific evidence suggests that the cargo of SEVs, comprising proteins, nucleic acids, and lipids, may be more relevant than merely their quantity or size in discerning between cancer and non-cancer patients. Their cargo could impact the signaling of the host’s immune defenses, facilitating the spread of the neoplasm by priming distant environments for metastasis [23,24]. This aspect is particularly crucial for identifying biomarkers and gaining a better understanding of cancer pathophysiology. Consequently, the number of studies focusing on the proteomics of SEVs in patients diagnosed with a mammary tumor has increased, with the potential for early diagnosis and longitudinal prognostic evaluation [9]. 

In this study, we observed that the protein characteristics of SEVs’ cargo can be more discriminatory than their mere presence or absence. The relative abundance of proteins among the various groups did not exhibit significant differences, rendering it an insignificant factor within the study’s context. However, the pertinent observation lies in the consistent detection of distinct proteins across all groups, despite certain groups sharing common proteins, presumably linked to normal cellular constituents of the species. This discernment has enabled the identification of potential biomarkers for diagnostic and prognostic purposes, and conceivably for guiding treatment modalities. 

The proteins identified in blood samples from female dogs, particularly in the malignant mammary tumor group, encompass various roles in cancer biology. One of these very abundant proteins, inter-alpha-trypsin inhibitor heavy chain (ITIH2), contributes to matrix stability and may serve as a hyaluronan transporter or binding protein [25,26]. These proteins can be positively or negatively regulated, but there is already evidence that all members of the ITIH family play a significant role in cell malignant processes and tumor growth [26]. Kopylov and colleagues (2020) [26] studied the proteomics of plasma samples from colorectal cancer patients at different stages and healthy patients. They observed significant changes in the levels of inter-alpha-trypsin inhibitor heavy chains (such as ITIH2) due to their implication in tumor growth and the malignancy process [26]. The literature describes that ITIH2 may present significant downregulation in solid tumors [27], contrary to the results obtained in the study by Kopylov and colleagues (2020) [26] and in our study, which observed a greater abundance of the ITIH2 protein in the malignant carcinoma group. According to Hamm (2008) [24], ITIH2 expression was clearly detectable in normal breast epithelium, in hyperplastic gland epithelium, and in ductal carcinoma in situ. However, in 44% of invasive breast carcinomas, ITIH2 expression was strongly reduced or completely lost, whereas 56% of invasive carcinomas maintained moderate to strong ITIH2 expression. Furthermore, a strong expression of ITIH2 was highly significantly associated with the presence of estrogen receptors. Our results may reflect differences between canine and human carcinomas, and further studies are needed to prove or refute these findings. Also, van den Broek and colleagues (2010) [27] observed significantly increased serum concentrations in the breast cancer group for ITIH4, significantly decreasing after surgery. The authors suggested the potential for ITIH4 in post-surgical breast cancer monitoring [27]. Accordingly, van Winden and colleagues (2010) [27] and Opstal-van Winden and colleagues (2011) [28] discovered that isoform 1 of the heavy chain of inter-alpha trypsin inhibitor H4 (ITIH4) presented elevated values in pre-diagnosis breast cancer and was already altered up to three years before cancer detection. Corroborating the previous studies, the study by Yang and colleagues (2016) [29] aimed to identify new serum peptide biomarkers for women with breast cancer (BC) through the analysis of the serum proteomic profile. Three peptide biomarkers were identified, including ITIH4, and the authors concluded that this protein has diagnostic and prognostic potential for breast cancer.

Another highly abundant protein found mostly in the malign carcinoma group was phosphopyruvate hydratase or alpha enolase (ENO1), which plays a crucial role in cancer development by promoting cell proliferation, invasion and metastasis. Its overexpression has been linked to chemoresistance and poor prognosis [30]. Song and colleagues (2015) [31] analyzed the involvement of ENO1 in tumor progression and the prognosis of human glioma. They concluded that an overexpression of ENO1 was associated with glioma progression, while reducing ENO1 expression led to a suppression of cell growth, migration and invasion progression by inactivating the PI3K/Akt pathway in glioma cells. In cancer metabolic reprogramming, ENO1 stimulates cancer cells to create energy largely through the breakdown of glucose in a non-oxidative way, rather than typical oxidative phosphorylation, which is known as the Warburg effect [32,33]. Indeed, mammary canine tumors undergo metabolic alterations [34] that favor tumor growth. Chu and colleagues (2011) [35] examined ENO1 expression by immunohistochemical staining and evaluated its importance in canine mammary carcinoma. ENO1 overexpression significantly correlated with shorter survival but was not associated with ER positivity in canine mammary carcinoma [35]. The study of Zamani-Ahmadmahmudi and colleagues (2014) [36] used serological proteomic analysis to detect autoantigens that provoke a humoral response in dogs with mammary tumors. They found four autoantigens, including ENO1, with significantly greater immunoreactivity in tumor samples than in normal samples, which were identified as biomarker candidates [36]. 

Besides ITIH2, ITIH4 and ENO1, we also found eukaryotic translation elongation factor 2 (eEF2) as an abundant protein. It is a member of the GTP-binding translation elongation factor family that is essential for protein synthesis. In our study, this protein was abundantly found in the SEVs’ proteomic of the malignant carcinoma group. Eukaryotic translation elongation factor 2 (eEF2) and its kinase eEF2K regulate many cellular processes, such as protein synthesis, cellular differentiation and malignant transformation [37]. Scientific evidence shows that eEF2K regulates the cell cycle, autophagy, apoptosis, angiogenesis, invasion and metastasis in various types of cancer. The expression of eEF2K promotes cancer cell survival, and the level of this protein is increased in many cancer cells to adapt them to microenvironmental conditions, including hypoxia, nutrient depletion, and acidosis [38]. The study by Wang and colleagues (2019) [39] investigated the role of autophagy and its regulator, eukaryotic elongation factor 2 kinase (eEF2K), in determining the biological nature of triple-negative breast cancers (TNBCs). eEF2K and autophagy play key roles in maintaining aggressive tumor behavior and chemoresistance in TNBC, and eEF2K silencing may be a new strategy for the treatment of TNBC.

Actin B (ACTB) was also found as abundant and exclusive protein in this study. Beta-actin (ACTB) is a cytoskeletal structural protein widely distributed in all eukaryotic cells and plays critical roles in cell migration and cell division. ACTB is generally upregulated in most tumor cells and tissues, being associated with the invasiveness and metastasis of cancers [40]. Fang et al. (2019) [41] evaluated the role of serum actin-4 as a biomarker for the diagnosis of BC, as well as the association between its levels and clinicopathological characteristics. They reported that the serum ACTN4 level was upregulated in BC patients compared to healthy controls [41]. Furthermore, high ACTN4 expression was significantly associated with clinical stage, tumor grade, and lymph node status [41,42]. Numerous clinical studies showed that changes in ACTN4 gene expression are correlated with aggressiveness, invasion, and metastasis in certain tumors [43]. Wang and colleagues (2017) [44] sought to determine the role and regulation of ACTN4 expression in human breast cancer metastasis under ellagic acid (EA)-based therapy. EA inhibited breast cancer growth and metastasis by directly targeting ACTN4 in vitro and in vivo. ACTN4 knockdown resulted in blocking malignant cell proliferation, colony formation and improving metastasis potency [44]. Increased ACTN4 expression was directly associated with advanced cancer stage, an increased incidence of metastases and a short survival period globally [44]. 

Chung and colleagues (2014) [45] used mass spectrometry to identify proteins differentially expressed in sera from women with breast cancer and healthy volunteers. From 51 protein peaks that were significantly up- or downregulated, they obtained 5 protein peaks that showed positive association with large tumor size and lymph node involvement and could distinguish women with breast cancer with high sensitivity and specificity. Interestingly, transthyretin and beta2-glycoprotein were between those proteins. In agreement, our study also identified an abundance and exclusive peak for both proteins. However, previously, the study by Nasim et al. (2012) [46] identified several acute phase proteins as potential biomarkers for breast carcinoma but reported that transthyretin was significantly downregulated in the sera of BC patients [46]. Recently, Sharma et al. (2023) [47] studied the action of organochlorine pesticides (OCPs) and their relationship with the spread of breast cancer in Indian women, through the proteomic analysis of plasma from breast cancer patients. They found 17 dysregulated proteins, but transthyretin (TTR) was three times higher than in healthy controls, and docking and molecular dynamics studies revealed a competitive affinity between the pesticide endosulfan II and the thyroxine-binding site of TTR that could result in endocrine dysregulation and lead to breast cancer. The authors pointed to a putative role of TTR in OCP-mediated CM [47]. 

Apolipoprotein H (APOH) or beta2-glycoprotein I (β2-GPI) is a plasma glycoprotein that has been implicated in a variety of physiological functions. Apolipoproteins (APOs) bind to lipids to form lipoproteins. By functioning as lipid carriers, apolipoproteins act as ligands for cell membrane receptors, enzyme cofactors and structural components of lipoproteins [48,49]. Several studies demonstrate the interaction of APOs with classical tumorigenesis pathways. Furthermore, the dysregulation of APOs may indicate the occurrence and progression of cancer, thus serving as potential biomarkers for cancer patients [50]. Besides Chung and colleagues [45], Lee and colleagues (2023) [51] also investigated the role of β2-GPI in tumor cells from mammary cancer patients and its correlation with tumor prognosis. β2-GPI expression was predominantly observed in the cells from breast cancer patients and significantly correlated with tumor stage and lymph node metastasis of breast cancer. A high expression of β2-GPI was significantly correlated with better overall survival (OS) and disease-free survival (DFS). Their results corroborated Lin and colleagues’ previously results [52]. 

Cytoskeletal rearrangement occurs in several cellular processes and involves a broad spectrum of proteins such as gelsolin superfamily proteins [53]. Gelsolin (GSN) is a multifunctional actin-binding protein that is greatly decreased in many transformed cell lines in tumor tissues, including breast cancers. The study by Mielnicki et al. (1999) [54] demonstrated epigenetic modification leading to a downregulation of gelsolin expression in human breast cancer. Baig et al. (2013) [53] investigated the germline mutations and expressional profile of gelsolin in human breast cancer tissues. Different types of mutations were observed in gelsolin coding regions, and the transcript level was significantly lower in breast tumor tissues compared to control samples, as well as in metastatic patients, compared to disease-free patients at final follow-up. The downregulation of gelsolin was also described by Winston and colleagues (2001) [55] in invasive mammary carcinomas. Stock et al. (2015) [56] analyzed gelsolin mRNA levels and concluded that high gelsolin levels are associated with a better prognosis in ER(+) HER2(−) breast cancer and a reduction in tumor cell migration. However, gelsolin was also found as exclusive protein in this study, instead of being downregulated. This finding was intriguing, but considering the differences between canine and human species and the scarcity of studies in dogs, we hypothesize that the abundance of gelsolin protein in samples from dogs with malignant mammary tumors may be related to a better prognosis in these individuals, not related to its function as a diagnostic biomarker. Therefore, it is likely that gelsolin may play a dual role in cancer genesis. For example, Rao and colleagues (2002) [57] reported decreased gelsolin expression in the early stages of malignant transformation in urothelial carcinomas. However, there was an increase in gelsolin expression in the transition from non-invasive to invasive urothelial carcinomas, indicating a biphasic expression pattern. Therefore, the authors concluded that increased gelsolin expression could indicate the conversion of a superficial tumor into an invasive tumor. Corroborating our finding, Van den Abbeele and colleagues (2007) [58] demonstrated that the downregulation of gelsolin in several cancer cell types significantly reduces the invasive and motile properties of the cells, as well as cell aggregation, pointing to a role for gelsolin as tumor activator. Also, Chen et al. (2015) [57] investigated the role of gelsolin (GSN) in epithelial-mesenchymal transition (EMT) in breast cancer cells in response to transforming growth factor beta 1 (TGF-β1). The results showed that TGF-β1 induced the expression of GSN and EMT and increased cell migration and invasion [59]. GSN overexpression affected cell proliferation, cell cycles and migration, and modulated vimentin expression [59]. The authors concluded that TGF-β1 promoted the demethylation of the GSN gene promoter in these cells, inducing epigenetic modifications that contributed to the progression of EMT in breast cancer cells [59]. In the study by Zhang et al. (2020) [60] the role of gelsolin (GSN) in hepatocellular carcinoma (HCC) was investigated. GSN was found to be overexpressed in HCC tissues and correlated with advanced tumor grade and poor prognosis, while GSN knockdown inhibited tumor cell migration and invasion and GSN overexpression had the opposite effect. The authors suggested that GSN promotes HCC progression by regulating epithelial-mesenchymal transition (EMT). 

Interestingly, vitamin D-binding protein (VDBP) or group-specific component (GC) was identified as an abundant protein in the remission group. We didn’t find literature about the relationship between cancer and VDBP, but Tagliabue and colleagues (2015) [61] performed a meta-analysis of vitamin D binding protein and cancer risk. They included 28 independent studies relating to the following tumors: basal cell carcinoma, bladder, breast, colon-rectum, endometrium, liver, esophagus, stomach, melanoma, pancreas, prostate and kidney. The authors concluded they could not assert that DBP would be a marker of cancer remission, but the results suggest a trend towards a decreased risk of cancer in individuals with higher levels of DBP, which could indicate a potential protective role of DBP in cancer.

The present study has several limitations that warrant attention and justification. Firstly, the absence of tissue samples from healthy animals precluded the performance of HE staining, limiting our ability to confirm healthy status and comprehensively evaluate biomarker expression variations through immunohistochemistry (IHC) staining. This limitation arose because we worked with dogs from veterinary clinics and, ethically, could not perform mammary tissue excision surgery without a clinical indication. Additionally, our study did not include phosphoproteome analysis, which could have provided additional information on the activity of various targets and enriched our current data. While we recognize the importance of the phosphoproteome in offering insights into target activity, our study focused on the general proteomic analysis of small extracellular vesicles (sEVs). Conducting a detailed analysis of phosphorylated proteins would require additional experiments and specific resources, which were beyond the initial scope of this project. We plan to incorporate phosphoproteome analysis in future studies to provide a more comprehensive understanding of the molecular mechanisms and signaling pathways involved in cancer biology.

Regarding the classification of extracellular vesicles (EVs), we excluded those larger than 150 nm following the MISEV2018 guidelines. Determining the precise origin of sEVs smaller than 150 nm remains challenging due to the lack of consensus on specific markers, necessitating advanced identification and functional assessment techniques. Finally, the lack of correlation with human data limits the direct applicability of our findings. Utilizing databases such as TCGA and the Human Protein Atlas for analysis would be essential to validate the relevance of the identified biomarkers in humans. These limitations can be addressed in future studies, expanding the scope of the research and strengthening the validity and applicability of the results to clinical practice.

## 5. Conclusions

Based on our main findings, the concentration and size of SEVs did not consistently vary in the canine mammary tumor (CMT) groups, before or after surgical treatment, or in the relapse, remission, and control groups. Therefore, SEVs did not prove to be reliable biomarkers and could not be strictly correlated with CMC diagnosis. However, in this descriptive study, the identification of candidate proteins for biomarkers underscores their pivotal role in defining diagnostic and prognostic strategies for CMTs. This achievement highlights the potential to enhance patient care and advocate for the well-being of our canine companions. The cargo of SEVs shows promise in predicting disease progression, motivating us to pursue further investigations for a comprehensive understanding. Importantly, there is limited literature on these proteins in dogs, and this study is the first to introduce them to the scientific community. This underscores the need for further investigations to comprehensively understand their potential in predicting disease progression and improving patient care for CMTs. 

## Figures and Tables

**Figure 1 cancers-16-02562-f001:**
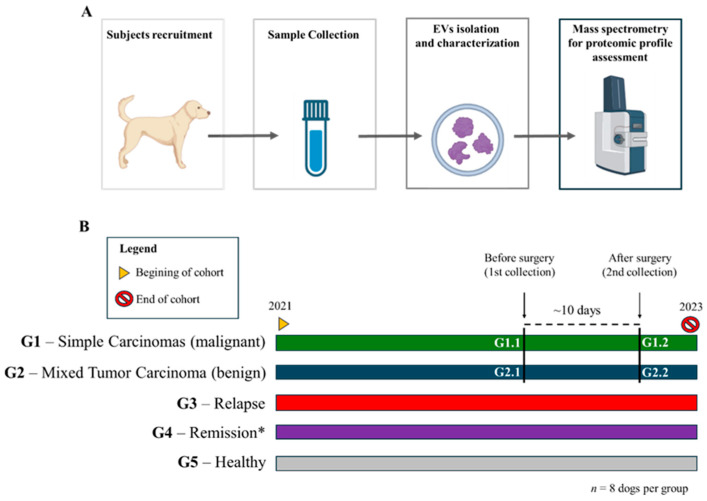
**Methodological Approach and Classification of Recruited Dogs**: (**A**) Overview of the methodological approach. Dogs with mammary tumors were recruited for the study. Samples were collected for the isolation and characterization of small extracellular vesicles (SEVs), followed by the assessment of their proteomic profiles using mass spectrometry. (**B**) Classification of recruited dogs into five groups: G1—simple carcinomas (malignant); G2—mixed tumor carcinomas (benign); G3—relapse of disease; G4—remission of disease; G5—healthy controls. Dogs in groups G1 and G2 had samples collected at two timepoints: G1.1 and G2.1—before any surgical intervention; G1.2 and G2.2—after mastectomy. The remission (G4) and relapse (G3) groups were treated as independent groups, with no paired comparisons. * Indicates statistical significance.

**Figure 2 cancers-16-02562-f002:**
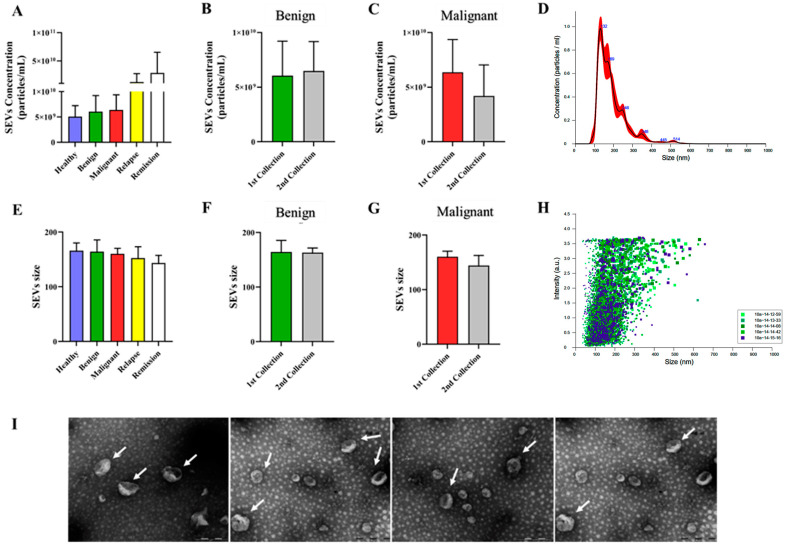
Characterization of SEVs. Concentration of SEVs isolated from plasma samples of dogs with mammary tumors (**A**) all groups without surgical intervention. Before and after mastectomy in (**B**) Benign and (**C**) Malignant tumors. (**D**) Representative Nanoparticle Tracking Analysis (NTA) profile showing SEVs concentration, with values displayed as particles/mL (*n* = 56). Size of SEVs in the respective groups: (**E**) All groups without surgical intervention, before and after mastectomy in (**F**) Benign and (**G**) Malignant tumors. (**H**) Representative NTA profile displaying size distribution, with values shown as arbitrary units (*n* = 56). (**I**) Transmission Electron Micrography (TEM) of SEVs isolated from plasma, magnified at 140,000×. White arrows indicated SEVs in the grids. Legend: SEVs—small extracellular vesicles.

**Figure 3 cancers-16-02562-f003:**
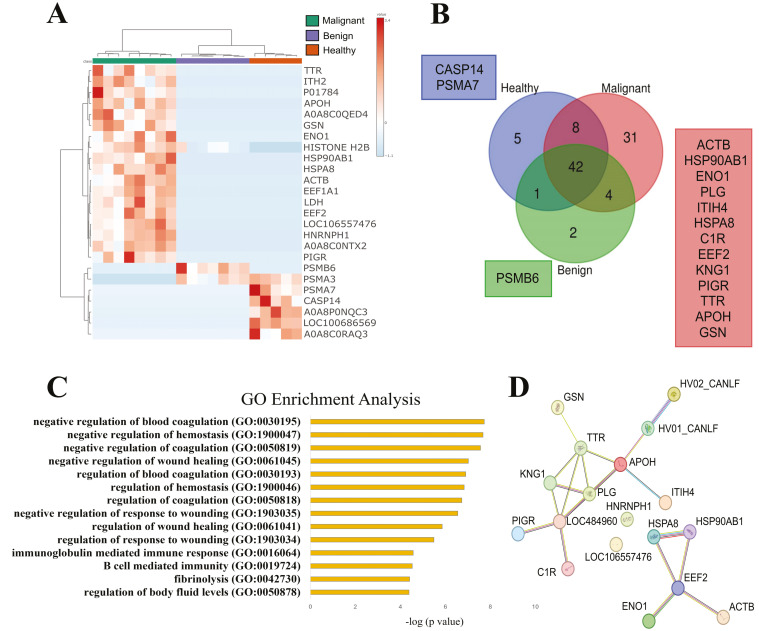
Proteomic profile of SEVs isolated from dogs with benign or malignant tumors compared to healthy controls. (**A**) Heatmap displaying the top 25 different proteins found in each condition. (**B**) Venn diagram illustrating the shared and unique proteins identified in benign, malignant, and healthy subjects. Ig-like or uncertain proteins retrieved from Uniprot were excluded. Squares denote the list of exclusive proteins. (**C**) Gene ontology enrichment analysis for the unique proteins identified in the malignant group. Only enrichments with *p* < 0.05 and FDR < 0.05 are reported, displayed a −log(*p* value). (**D**) Molecular interactions potentially involved in the unique proteins exclusively found in the malignant condition, retrieved from STRING database.

**Figure 4 cancers-16-02562-f004:**
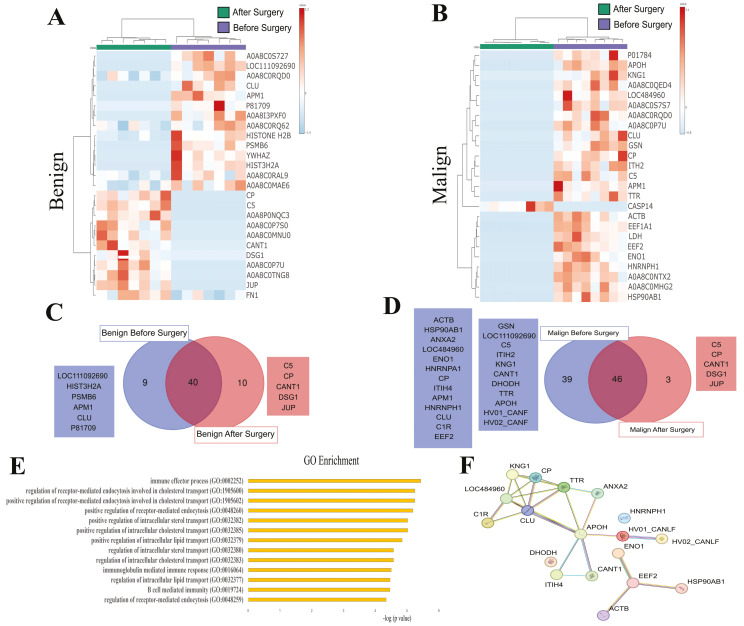
Proteomic profile of SEVs isolated from dogs with benign or malignant mammary tumors before and after surgery. (**A**) Heatmap displaying the top 25 different proteins found in benign condition. (**B**) Heatmap displaying the top 25 different proteins found in malignant condition. (**C**) Venn diagram illustrating the shared and unique proteins identified in benign tumors. (**D**) Venn diagram illustrating the shared and unique proteins identified in malignant tumors before and after mastectomy. Ig-like or uncertain proteins retrieved from Uniprot were excluded. Squares denote the list of exclusive proteins. (**E**) Gene ontology enrichment analysis for the unique proteins found in the malignant group. Only enrichments with *p* < 0.05 and FDR < 0.05 are reported, displayed as −log (*p* value). (**F**) Molecular interactions potentially involved in the unique proteins exclusively found in the malignant condition, retrieved from STRING database.

**Figure 5 cancers-16-02562-f005:**
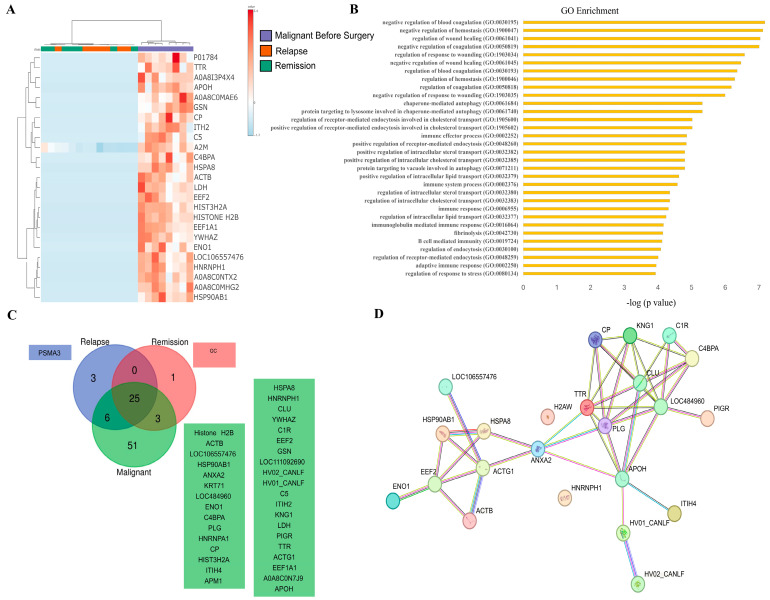
Proteomic profile of SEVs isolated from dogs with malignant mammary tumors compared to SEVs from remission and relapse condition. (**A**) Heatmap displaying the top 25 different proteins found in each condition. (**B**) Venn diagram illustrating the shared and unique proteins identified. Ig-like or uncertain proteins retrieved from Uniprot were excluded. Squares denote the list of exclusive proteins. (**C**) Gene ontology enrichment analysis for the unique proteins found in the malignant group. Only enrichments with *p* < 0.05 and FDR < 0.05 are reported, displayed as −log (*p* value). (**D**) Molecular interactions potentially involved in the unique proteins exclusively found in the malignant condition, retrieved from STRING database. Neither remission nor relapse condition showed set enrichment.

**Figure 6 cancers-16-02562-f006:**
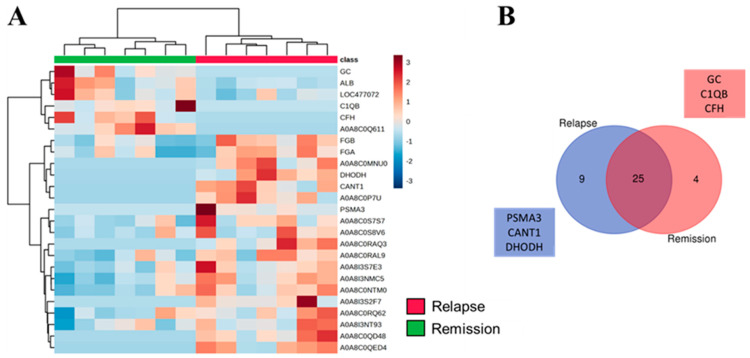
Comparison of the SEVs proteomic profile isolated from dogs that underwent remission or relapse. (**A**) Heatmap displaying the top 25 different proteins found in each condition. (**B**) Venn diagram illustrating the shared and unique proteins identified. Ig-like or uncertain proteins retrieved from Uniprot were excluded. Squares denote the list of exclusive proteins. Neither remission nor relapse showed set enrichment.

## Data Availability

To ensure transparency and data accessibility, the raw mass spectrometry files, FASTA files of protein sequences, compressed files containing MaxQuant ‘/combined/txt/’ outputs, and any other relevant data will be deposited in a public online proteomics repository such as ProteomeXchange. Additionally, a comprehensive table mapping sample-level, LC-MS, and MaxQuant files (see Appendix A) will be provided, facilitating transparency and reproducibility of our analyses.

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
