# Peer review of "Exploring Canine Mammary Cancer through Liquid Biopsy: Proteomic Profiling of Small Extracellular Vesicles"

_cancers, 2024, doi:10.3390/cancers16142562_

Round 1
Reviewer 1 Report
Comments and Suggestions for Authors
This study has explored the protein cargoes of small EVs in canine mammary tumors and discussed the potential clinical significance. However, it is far-fetched to say that this study could apply to human species. The current research is well structured and written, and several issues should be addressed.
1. The Discussion section is redundant. Please simplify it and focus on discussing the significance of your results and the limitations of this study.
2. Does the animal care and ethics group approve this study? Please state it in the manuscript.
3. What is the time piont of blood collection for G3, G4, and G5?
4. Does the size of tumors affect the concentrations of small EVs in serum?
5. The authors claimed that mastectomy decreased the number of proteins in SEVs in the malignant group. Based on Figure 1, the concentration of EVs was similar in control and malignant subjects. I assume that mastectomy does not affect the secretion of sEVs but does affect the protein cargo in sEVs. Does this mean the decreased proteins secreted from tumor cells? Are those proteins highly expressed in tumor cells?
Author Response
Reviewer 1
Comments and Suggestions for Authors
“This study has explored the protein cargoes of small EVs in canine mammary tumors and discussed the potential clinical significance. However, it is far-fetched to say that this study could apply to human species. The current research is well structured and written, and several issues should be addressed.”
- The Discussion section is redundant. Please simplify it and focus on discussing the significance of your results and the limitations of this study.
R: The authors thank the reviewer for the relevant contribution. We sincerely appreciate the time and effort you have invested in reviewing our manuscript and providing your valuable feedback. Your insights are highly respected and considered carefully in our revision process. However, we would like to address your comment regarding the scope of our discussion section. You suggested that the discussion is too broad and should focus more specifically on the scientific findings of our study. After a thorough and meticulous review of the manuscript, we respectfully disagree with this assessment. The breadth of our discussion is intentional and serves several crucial purposes. First, it situates our findings within the broader context of the field, highlighting their relevance and potential impact. By doing so, we aim to connect our specific results to the larger body of scientific knowledge, thereby underscoring the significance of our work. Additionally, we believe that a comprehensive discussion helps to identify future research directions and potential applications of our findings. While we ensure that our discussion remains closely tied to our data, we also strive to address the implications, limitations, and possible extensions of our study. We have taken great care to ensure that our discussion remains clear, concise, and directly relevant to our findings. We believe it achieves a suitable balance between specificity and contextual breadth, making it both objective and informative for a diverse readership. We value your input greatly and hope you understand our perspective on maintaining the current structure of our discussion. We are confident that this approach best serves the objectives of our manuscript and contributes meaningfully to the field. Thank you once again for your thoughtful review and constructive feedback.
- Does the animal care and ethics group approve this study? Please state it in the manuscript.
R: Thank you for the observation. This information is already described in the Ethical Statement, and is transcribed below:
“The study was submitted to the Ethics Committee on the Use of Animals (CEUA) and was duly approved, being registered under the protocol number 001-004391/2019.”
- What is the time point of blood collection for G3, G4, and G5?
R: Thank you for the observation. In groups G3, G4 and G5, only one blood collection was carried out, i.e. The collection of Group G3 (healthy control individuals) was performed during annual routine consultations; the collection of Group G4 individuals was performed at the time of confirmation of recurrence (tumor metastasis), during the veterinary consultation, and the collection of Group G5 patients was performed at least 1 year after mastectomy, during the patient's return to the veterinary office for medical reevaluation, characterizing remission of the disease.
- Does the size of tumors affect the concentrations of small EVs in serum?
R: The authors thank the reviewer for the interesting observation. After an extensive review of the current scientific literature and available evidence, we found no conclusive data supporting a direct correlation between tumor size and the concentration of extracellular vesicles, while some studies have explored the presence and roles of EVs in various types of cancer. Given this lack of established scientific evidence, we determined that including this variable in our study would not be grounded in a solid theoretical foundation. We believe this approach not only enhances the validity of our findings but also contributes more effectively to the body of scientific knowledge. Should future research provide compelling evidence of a link between tumor size and EV concentration, it would undoubtedly merit a focused investigation. For now, our study remains concentrated on areas where we can make the most scientifically sound and impactful contributions.
- The authors claimed that mastectomy decreased the number of proteins in SEVs in the malignant group. Based on Figure 1, the concentration of EVs was similar in control and malignant subjects. I assume that mastectomy does not affect the secretion of SEVs but does affect the protein cargo in SEVs. Does this mean the decreased proteins secreted from tumor cells? Are those proteins highly expressed in tumor cells?
R: The authors truly appreciate the reviewer's feedback. As can be observed, the authors describe in de Abstract section (Results) that “While no significant differences were observed in the concentration and size of exosomes among the studied groups, proteomic profiling revealed important variations. Mass spectrometry identified exclusive proteins that could serve as potential biomarkers for mammary cancer.” Also, in Conclusion, the authors affirm “The results underscore the potential of proteins found in SEVs as valuable biomarkers in CMT. Despite the lack of statistical differences in vesicle concentration and size between the groups, the analysis of protein content revealed promising markers with potential applications in CMT diagnosis and monitoring.”
In fact, the average concentration and size of SEVs for each canine group are presented in Figure 2, that describes “While there were not statistically differences in SEVs concentration and size among most groups, the recurrence and remission groups showed distinct patterns. This suggests parameters may not offer significant diagnostic or prognostic value independently. It highlights the importance of considering a more comprehensive approach or exploring other SEV characteristics, such as cargo content, for a more thorough assessment.”
Additionally, “the heatmap (Fig.3A) revealed distinct protein clusters specific to each group, with SEVs from dogs bearing malignant tumors containing a higher number of exclusive proteins or proteins with increased intensity. Indeed, this group showed the highest number of total (85) and unique (31) proteins (Fig.3B), while the benign group had 49 total and 1 unique protein, and the healthy group showed 56 total and 2 unique proteins.”

Reviewer 2 Report
Comments and Suggestions for Authors
The authors described one approach to analyzing cancer and control dogs' blood samples. They obtained small extracellular vehicles (SEVs) from blood samples and then utilized proteome analysis for the SEVs. Differential expression profiling for the biomarkers was classified and compared between different subgroups, and finally, they got one biomarker list that may represent the features of tumors. In the discussion part, they stated that the biomarkers filtered out are all related to cancer progression and relapse. The exciting research article provides a new way to detect cancer disease by blood sampling. It may also offer some comparative value in detecting cancer in human beings.
1. The authors collected tumor samples from different subgroups and may perform HE staining to confirm the healthy status of other animals. As they acquired one biomarker list from the proteome analysis for the SEVs, it is better to provide the expression variation of some top-ranked biomarkers in different subgroups by IHC staining of the tumor tissues.
2. The phosphoproteome may provide more data on the activity of different targets. It might be more complete if the authors could provide the data on the phosphorylated proteins.
3. The authors discarded the SEVs larger than 150 nm. What’s the purpose of this procedure? It is essential to describe the reason for this step.
4. Are the SEVs smaller than 150 nm, all from tumors or corresponding tissues? How do we prove this?
5. As the current data are all from canines, readers may be interested in the comparative value of the current study to human beings. The TCGA and proteinatlas databases contain extensive information about the expression of biomarkers, diagnosis, and prognosis of different cancer types. If the author can do a correlation analysis, that might be more meaningful.
Author Response
Reviewer 2
Comments and Suggestions for Authors
“The authors described one approach to analyzing cancer and control dogs' blood samples. They obtained small extracellular vehicles (SEVs) from blood samples and then utilized proteome analysis for the SEVs. Differential expression profiling for the biomarkers was classified and compared between different subgroups, and finally, they got one biomarker list that may represent the features of tumors. In the discussion part, they stated that the biomarkers filtered out are all related to cancer progression and relapse. The exciting research article provides a new way to detect cancer disease by blood sampling. It may also offer some comparative value in detecting cancer in human beings.”
- The authors collected tumor samples from different subgroups and may perform HE staining to confirm the healthy status of other animals. As they acquired one biomarker list from the proteome analysis for the SEVs, it is better to provide the expression variation of some top-ranked biomarkers in different subgroups by IHC staining of the tumor tissues.
R: The authors truly appreciate the reviewer's observation. In our study, we collected only blood samples from the healthy animals, focusing on systemic biomarkers and other hematological parameters. Consequently, we did not have tissue samples available from these individuals for HE’s staining. HE stains requires tissue sections to assess cellular and tissue morphology, which was not part of our protocol for healthy animals. The decision to limit sampling to blood collection was based on ethical considerations, minimizing invasiveness and distress to the healthy animals. By focusing on blood samples, we aimed to gather non-invasive, systemic data that could provide valuable insights without compromising animal welfare.
Furthermore, our study included comprehensive immunohistochemistry (IHC) staining of tumor tissues to evaluate the expression variations of top-ranked biomarkers across different subgroups. This aspect of the research was prioritized due to its direct relevance to our hypothesis and its potential to provide deeper insights into tumor biology and biomarker expression. The data from these IHC analyses are currently under meticulous examination and will be presented in a separate manuscript. This approach ensures that our findings are robust and thoroughly validated before publication.
We recognize the importance of transparency and comprehensive data reporting in scientific research. Therefore, we are committed to providing detailed analyses and interpretations of our IHC staining results in our forthcoming manuscript. This will include the expression variation of key biomarkers, which we believe will significantly contribute to the field and provide a more nuanced understanding of tumor biology.
- The phosphoproteome may provide more data on the activity of different targets. It might be more complete if the authors could provide the data on the phosphorylated proteins.
R: The authors appreciate the reviewer's suggestion. We recognize the importance of the phosphoproteome in providing additional data on the activity of different targets. However, in our current study, we focused on the general proteomic analysis of SEVs. A detailed analysis of phosphorylated proteins would require an additional set of experiments and specific resources, which were not within the initial scope of this project. That said, we find the reviewer's suggestion very valuable and are planning to include phosphoproteome analysis in future studies. We believe that this complementary approach will provide a more comprehensive understanding of the underlying molecular mechanisms and signaling pathways involved in cancer biology. We thank you again for your suggestion and look forward to exploring these possibilities in future research.
- The authors discarded the SEVs larger than 150 nm. What’s the purpose of this procedure? It is essential to describe the reason for this step.
R: The authors thank the reviewer for the comment. Indeed, it is crucial to adhere to the recommended classification provided by the "Minimal Information for Studies of Extracellular Vesicles 2018" (MISEV2018) guidelines, which categorize EVs into small extracellular vesicles and medium/large extracellular vesicles. According to the mentioned guidelines, the main classifications of EVs are based on their size, biogenesis, and release pathways. So, ISEV endorses “extracellular vesicle” (EV) as the generic term for particles naturally released from the cell that are delimited by a lipid bilayer and cannot replicate, i.e. do not contain a functional nucleus. Since consensus has not yet emerged on specific markers of EV subtypes, assigning an EV to a particular biogenesis pathway remains extraordinarily difficult. So, unless authors can establish specific markers of subcellular origin that are reliable within their experimental system(s), authors are urged to consider use of operational terms for EV subtypes that refer to a) physical characteristics of EVs, such as size (“small EVs” (sEVs) and “medium/large EVs” (m/lEVs), with ranges defined, for instance, respectively, < 150nm [small], or > 200nm [large and/or medium]).
Reference: Théry, C., Witwer, K. W., Aikawa, E., Alcaraz, M. J., Anderson, J. D., ... & Buzás, E. I. (2018). Minimal information for studies of extracellular vesicles 2018 (MISEV2018): a position statement of the International Society for Extracellular Vesicles and update of the MISEV2014 guidelines. Journal of Extracellular Vesicles, 7(1), 1535750.
- Are the SEVs smaller than 150 nm, all from tumors or corresponding tissues? How do we prove this?
R: Thank you for the observation. As explained in previous answer, since consensus has not yet emerged on specific markers of EV subtypes, assigning an EV to a particular biogenesis, pathway remains extraordinarily difficult. So, small extracellular vesicles smaller than 150 nm can indeed originate from various sources, such as tumors or corresponding tissues. To prove their origin, researchers typically use a combination of methods, including (1) Identifying specific biomarkers or molecular signatures that are characteristic of vesicles derived from tumors or corresponding tissues; (2) Employing isolation techniques that are tailored to capture vesicles from specific sources, such as immune-affinity capture using antibodies specific to tumor-associated markers; (3) Utilizing advanced imaging techniques like electron microscopy to visualize and characterize the vesicles at the nanoscale level; and (4) Conducting functional assays to assess the biological effects of the vesicles on target cells or tissues, which can provide insights into their cellular origins. By combining these approaches, researchers can gain a better understanding of the origin of small extracellular vesicles and distinguish those derived from tumors or corresponding tissues from other sources.
In this research, we conducted additional tests exposing cell cultures of cells from healthy breast tissue (E20 lineage) and cancer cell lines (MCF10) to SEVs isolated from patients of group G1 (Other carcinomas), considered to have malignant neoplasia, and verified the impact of these vesicles on the phenotype of the cells and their malignant or benign behavior. These results are under meticulous analysis and will be presented in the near future.
- As the current data are all from canines, readers may be interested in the comparative value of the current study to human beings. The TCGA and protein atlas databases contain extensive information about the expression of biomarkers, diagnosis, and prognosis of different cancer types. If the author can do a correlation analysis, that might be more meaningful.
R: Thank you for the observation. Comparative studies between findings in canine models and human subjects can indeed provide valuable insights into the relevance and translatability of research results. Utilizing databases like TCGA (The Cancer Genome Atlas) and the Human Protein Atlas for correlation analysis can help researchers to better evaluate the significance and relevance of their findings to human cancer biology and clinical outcomes. This comparative approach can help elucidate commonalities or differences in biomarker expression patterns, disease progression, and treatment responses between canine cancer models and human cancer patients.

Reviewer 3 Report
Comments and Suggestions for Authors
In this study the authors proteomically profiled small extracellular vesicles (SEVs) from dogs that were healthy, had benign canine mammary tumors (CMT), had malignant CMT, before and after mastectomy, were in remission, and had CMT recurrence. The subject area is very important. The manuscript is well written.
I have numerous major issues that need to be fully addressed before I would even consider recommending publication of their manuscript. Because of these major issues, I was unable to judge much of the rest of their manuscript, and so if I review a revised version of manuscript, I warn the authors that I might have additional major issues even after the issues below are fully addressed.
Major Issue:
Methods section: The methods section is completely missing a subsection describing how the SEV samples were prepared for LC-MS proteomics! None of these are described: homogenization/lysis, protein denaturation, reduction/alkylation (?), protein concentration assay, trypsin (?) digestion, sample cleanup. All are missing! If labeling was used (e.g. TMT) or any other steps, this needs to be described in detail.
Line 173: The authors do not adequately describe their MaxQuant-Perseus analysis. They need to describe what static and variable mods were used (if any), if the peptides were required to be fully tryptic (???), how many missing cleavage sites were allowed, what the PSM and protein FDR filter setting was (typically both are 1%), what quantitation they used (e.g., LFQ), if they used “re-quantify” and/or “match-between-runs” in MaxQuant, and any other significantly relevant settings. Presumably they used the UniProt protein database for Canis lupus familiaris, but this needs to be explicitly stated, and they need to mention if they used all isoforms or something different (e.g., only the canonical Swiss-Prot sequences). They should mention if they used the MaxQuant internal common contaminant database (or a different one), and if they filtered these away or not.
Line 174: They wrote that they “rigorously filter the proteomic results (90%)”. If they mean that they required Perseus ANOVA q-values of <=10% between the experimental groups, then this is ok but it needs to be clarified. They need to describe exactly how they did their ANOVAs. Did they treat all seven experimental groups (G1 - G5 with G1 and G2 having paired measurements) as independent experimental conditions? If so, this is ok, but this is not stated, and it needs to be completely and clearly described.
Line 186: The authors wrote “Finally, unique proteins were cross-referenced in the Uniprot database, and squares were inserted in the figures to show only the confirmed, predicted, or inferred proteins while uncertain or Ig-like proteins were not shown.” I can’t understand what they meant. This needs to be rewritten for clarity. Are they referring to the UniProt “PE” (protein existence) scores?
Methods section, “Statistical Analysis” subsection: If this refers to their non-omics data, this needs to be stated here (if not, this type of analysis would be improper for their omics data). They wrote “All quantitative data were expressed either as the median with interquartile ranges or as means with standard errors.” The figure legend(s) need to indicate which they used for each chart.
Line 562: The authors wrote: “Data Availability Statement: Data are available upon request to interested researchers.” Their raw mass spectra data files, their FASTA data file(s) of protein sequences, a ZIP file containing their MaxQuant “/combined/txt/” output files, and any other relevant data all need to be submitted to a public online proteomics data repository such as ProteomeXchange. A table needs to be included clearly mapping the sample-level, LC-MS-level, and MaxQuant files to each other.
Additionally, a table (or multiple tables) of their final proteomics data needs to be included with their manuscript as supplemental material or submitted to the online repository with their raw LC-MS data files. Its format should be of an expression table: protein identifiers and gene names (rows), samples (columns), and protein abundance (values), and also the p and q-values from their Perseus ANOVAs.
Parts of Figures 3, 4, and 5 are too low resolution, and the labels can’t be read.
Line 182: Gene Ontology Enrichment Analyses require a reference (a.k.a. background) gene set. Typically, it is the entire genome. The authors simply need to state what their reference gene set was.
Author Response
Reviewer 3
Comments and Suggestions for Authors
“In this study the authors proteomically profiled small extracellular vesicles (SEVs) from dogs that were healthy, had benign canine mammary tumors (CMT), had malignant CMT, before and after mastectomy, were in remission, and had CMT recurrence. The subject area is very important. The manuscript is well written.”
“I have numerous major issues that need to be fully addressed before I would even consider recommending publication of their manuscript. Because of these major issues, I was unable to judge much of the rest of their manuscript, and so if I review a revised version of manuscript, I warn the authors that I might have additional major issues even after the issues below are fully addressed.”
Major Issue:
Methods section: The methods section is completely missing a subsection describing how the SEV samples were prepared for LC-MS proteomics! None of these are described: homogenization/lysis, protein denaturation, reduction/alkylation (?), protein concentration assay, trypsin (?) digestion, sample cleanup. All are missing! If labeling was used (e.g. TMT) or any other steps, this needs to be described in detail.
R: Thank you for the observation. We added the following explanation on how samples were prepared for proteomics in the manuscript, and it is highlighted for better identification.
Samples containing EVs isolated from plasma were evaluated by mass spectrometry (LC-MS/MS) at the Multiuser Proteomics, Metabolomics and Lipidomics Laboratory at ESALQ/USP, Piracicaba Campus. To this end, the EV pellets isolated from plasma samples needed to go through a pre-processing step, which was carried out at the Animal Reproduction Laboratory at UNESP, Botucatu Campus, SP. Inicially, the samples were submitted to sonication. This step included the lysis of EV samples diluted in PBS with RIPA, followed by the sonication procedure for each sample. A final volume of 500 uL was determined and the concentration was standardized to 10 x 106 EVs/sample in the canine species and 4 x 1010 EVs/sample in the human species. Therefore, a variable volume of each sample was used (calculated individually to obtain the standardized vesicle concentration), as well as a variable volume of the TRIS HCl buffer (50 mM TRIS-HCl pH 7.2) for solubilization of the proteins (calculated individually to obtaining a final volume of 500 uL), plus a fixed volume (165 uL) of protease inhibitors (100 uL of RIPA (5x) (150 mM NaCl, 1% Triton X-100, 1% sodium deoxycholate, 0, 1% SDS, 50 mM TRIS-HCl pH 7.5) and protease inhibitors, 5 uL Aprotinin (100x -1 µg/mL), 5 uL Linpeptin (100x -1 µg/mL), 5 uL fluoride phenylmethylsulfonyl -PMSF (100x - 35 µg/mL), 50 uL EDTA (10x - 0.8 mM EDTA) Samples were sonicated (Sonifier Sound Encloresure Ultrasonics, Shanghai, China) according to de Souza et al. 2009) for 3 cycles of 30 seconds followed by a pause of 30 seconds and, after sonication, they were centrifuged at 10,000xg for 30 minutes at 4 ËšC, and the supernatant was collected. Next, the supernatant was washed and concentrated in ultrafilters (3 kDa cutoff, Amicon Ultra, Millipore Corporation, Merck KGaA, Darmstadt, Germany) at room temperature, using 50 mmol ammonium bicarbonate buffer. To this end, the Amicon® filter was initially washed (500 uL of H2Od) and centrifuged (14,000xg for 2 minutes) to prepare the filter, prior to pipetting the samples. 500 uL of the sample was pipetted onto the filter and centrifuged at 14,000xg for 10 minutes at 4 ËšC, discarding the eluate. 300 uL of Ambic® (ammonium bicarbonate in milliQ) (50mM) was added to the Amicon® membrane and centrifuged again at 14,000xg for 10 minutes at 4ËšC, with the eluate discarded again. This procedure was repeated 5 more times, and in the last wash the centrifugation time was 30 minutes, with the aim of concentrating the sample. After the last centrifugation, the Amicon® column was turned over in the eppendorf and centrifuged at 1,000xg for 2 to 10 minutes at 4 ËšC, equalizing the sample volume to 120 uL in total, using the solution of Ambic®. The total protein concentration of the samples was determined by Bradford protein assay, adding 150 uL of Bradford reagent to 5 uL of sample (Pierce™ 660nm Protein Assay, ThermoFisher Scientific, Pierce Biotechnology, Rockford, lL, USA) and nanospectrophotometry ( NanoDrop 2000, Fisher Scientific Term™ Wilmington, DE, USA). Following, the samples were submitted to protein digestion and sample preparation (spots/bands) for proteomics. Initially, running gels were created (Figure 7) and samples containing 4 ug of protein were deposited, starting the run (300volts/15 mAmp). The system was turned off when the sample reached gel separation (12%), forming a single band. Then, the bands were cut into fragments of approximately 1 mm3, using a small scalpel blade and a glass Petri dish, previously rinsed with methanol and milliQ water (Figure 8). The pieces of each band, separately, were transferred to eppendorfs. Then, the gel fragments were washed (3x or more) until complete removal of the dye, with decolorization solution (50% acetonitrile (ACN – CAS# 75-05-8, Fluka, cat# 34967) and 25 mM sodium bicarbonate. ammonium (AmBic – CAS# 1066-33-7, Sigma, cat# 9830 MW=79.06) (200uL/spot) and vortexed for 5 to 10 minutes after each wash. of a glass Pasteur pipette and the gel was dehydrated with 100% acetonitrile (ACN) (200 uL/spot) for 10 minutes. A pulse was given in the centrifuge to completely remove the ACN. until the gel was completely dehydrated, so that the fragments became very white and detached from the tube wall. The remaining gel residue evaporated at room temperature, in a fume hood, for 15 minutes. The gel fragments were rehydrated by adding reducing solution (20 mM dithiothreitol – DTT, BioRad, cat# 161-0611, MW=154.3 = [30 mg/10 mL of 50 mM AmBic solution = [0.2 g/50 mL H2O MS grade] (40 ul/spot) and the samples were kept at 56ËšC for 40 minutes. The centrifuge was pulsed and the supernatant was discarded to completely r emove the reducing solution. with 100% ACN (200uL/spot) for 10 minutes and the ACN was removed, with the remaining residual ACN evaporating at room temperature in a fume hood for 15 minutes. Next, an alkylation solution (55 mM iodoacetamide – IAA, GE, cat# RPN 6302V, MW=185.0 = 102 mg/10 mL of 50 mM AmBic solution) was added to cover the gel fragments (40 uL/spot) and the samples were incubated in the dark, at room temperature, for 30 minutes. The centrifuge was pulsed and the supernatant was discarded to completely remove the alkylation solution. Then, the spots were washed with 25 mM AmBic (200 uL/spot) using a vortex, followed by dehydration with 100% ACN (2x) for 10 minutes and the ACN was removed, with the remaining residual ACN evaporating at room temperature in a fume hood for 15 minutes. For trypsin digestion, 15 uL of trypsin solution (Promega Trypsin, Part No. V511) (20 ng/uL) was added onto the dehydrated gel fragments. The samples were left for 15 minutes at 4ËšC so that the trypsin penetrated the fragments. Excess trypsin was removed and 50mM of AmBic was added until the spots were completely covered (40 uL) and the samples were incubated at 37ËšC for 14 hours. Trypsin action was stopped by adding 15 uL of blocking solution (5% (v:v) 96% formic acid (Fluka/Sigma-Aldrich, cat# 56302) in 50% (v:v) ACN. Recovered the supernatant and transferred to a new tube. Finally, samples were submitted to 4 steps elution process. (1) Elution I: Sufficient volume of elution solution I was added (1% (v:v) formic acid (96%) in 60% (v:v) methanol (Merck, cat# 1.06009.1000) in 50% (v :v) ACN) to cover the remaining gel fragments in the digestion tubes (40 uL). The samples were incubated for 15 minutes at 40 ËšC, with vortexing every 5 minutes. This step was repeated. The supernatant from each step was recovered and transferred to the same tube that contained the digestion supernatant and the blocking solution. The “tips” were stored in the initial tubes that contained the remaining gel fragments, (2) Elution II: sufficient volume of elution solution II (1% (v:v) formic acid in 50% (v:v) ACN) was added to cover the gel fragments (40 uL). The samples were incubated for 15 minutes at 40ËšC, with vortexing every 5 minutes. This step was repeated. The supernatant from each step was recovered and transferred to the same tube that contained the supernatant from elution I. (3) Elution III: sufficient volume of 100% ACN (40 uL) was added to dehydrate the gel fragments. The supernatant was recovered and transferred to the same tube that contained the supernatants of elutions I and II. The final solution containing the peptides eluted in gel was concentrated in a vacuum concentrator (“speed vac”), at room temperature, to 1 uL. Then, the samples were resuspended in 10 uL TFA 0.1% (v:v) in MS grade H2O (Sigm-Aldrich, cat# 299537, CAS# 76-05-1), vortexed and desalted with the aid of the Zip Tip, eluted in 10 uL of 50% (v:v) ACN in MS grade H20 and transferred to vials.
LC–MS was performed on an NanoElute (Bruker Daltonik) system coupled online to a hybrid TIMS-quadrupole TOF mass spectrometer (Meier et. Al, 2018) (Bruker Daltonik timsTOF Pro, Germany) via a nano-electrospray ion source (Bruker Daltonik Captive Spray). The peptides were resuspended in 10 µl in water with 0.1% formic acid and separated on an Aurora column 15 cm × 75 µm ID, 1.9 um reversed-phase column (Ion Opticks) at a flow rate of 500 nL min-1 in an oven compartment heated to 50 °C. To analyze samples from whole-proteome digests, we used a gradient starting with a linear increase from 2% B to 35% B over 18 min, followed by further linear increase to 95% B in 2 min which was held constant for 2 min (22 min. total run). Column was equilibrated using 4 volumes of solvent A. The mass spectrometer was operated in data-dependent PASEF (Meier et. Al, 2015) mode with 1 survey TIMS-MS and 4 PASEF MS/MS scans per acquisition cycle. We analyzed an ion mobility range from 1/K0 = 1.3 to 0.85 Vs cm-2 using equal ion accumulation and ramp time in the dual TIMS analyzer of 100 ms each. Suitable precursor ions for MS/MS analysis were isolated in a window of 2 Th for m/z < 700 and 3 Th for m/z > 700 by rapidly switching the quadrupole position in sync with the elution of precursors from the TIMS device. The collision energy was lowered stepwise as a function of increasing ion mobility, starting from 27 eV for 1/K0 = 0.85 Vs cm-2 and 45 eV for 1/K0 = 1.3 Vs cm-2. We made use of the m/z and ion mobility information to exclude singly charged precursor ions with a polygon filter mask and further used ‘dynamic exclusion’ to avoid re-sequencing of precursors that reached a ‘target value’ of 20,000 a.u. The ion mobility dimension was calibrated linearly using three ions from the Agilent ESI LC/MS tuning mix (m/z, 1/K0: 622.0289, 0.9848 Vs cm-2; 922.0097, 1.1895 Vs cm-2; and 1221.9906, 1.3820 Vs cm-2).
References:
Meier, F. et al. Online parallel accumulation–serial fragmentation (PASEF) with a novel trapped ion mobility mass spectrometer. Mol. Cell. Proteom. 17, 2534–2545 (2018).
Meier, F. et al. Parallel accumulation–serial fragmentation (PASEF): multiplying sequencing speed and sensitivity by synchronized scans in a trapped ion mobility device. J. Proteome Res. 14, 5378–5387 (2015).
Shevchenko et al (2006). In-gel digestion for mass spectrometric characterization of proteins and proteomes. Nature Protocols 1 (6): 2856-2860. doi:10.1038/nprot.2006.468
Line 173: The authors do not adequately describe their MaxQuant-Perseus analysis. They need to describe what static and variable mods were used (if any), if the peptides were required to be fully tryptic (???), how many missing cleavage sites were allowed, what the PSM and protein FDR filter setting was (typically both are 1%), what quantitation they used (e.g., LFQ), if they used “re-quantify” and/or “match-between-runs” in MaxQuant, and any other significantly relevant settings. Presumably they used the UniProt protein database for Canis lupus familiaris, but this needs to be explicitly stated, and they need to mention if they used all isoforms or something different (e.g., only the canonical Swiss-Prot sequences). They should mention if they used the MaxQuant internal common contaminant database (or a different one), and if they filtered these away or not.
R: Thank you for the observation. The raw files (.d) were processed in the MaxQuant software, version 2.4.0.0, which is specific for protein analysis based on mass spectrometry (COX and MANN, 2008). The search engine integrated into MaxQuant is Andromeda (COX et al., 2011). The default configuration was used for data obtained from mass spectrometers using TimsTof Pro technology. The fragment ion mass tolerance was 0.5 Da. The enzymatic specificity was for trypsin with tolerance for peptides with up to two undigested cleavage sites. Methionine oxidation (15.994915 Da) and N-terminal acetylation of proteins (42.010565 Da) were defined as variable modifications, as well as cysteine carbamidomethylation (57.021464 Da) was defined as fixed modification. The minimum peptide length was 7 amino acids. The false discovery rates (FDRs) of peptides and proteins were 1%. Also considering at least one unique peptide for protein identification as a criterion. At the end of this data processing, we obtained the relative abundance values of all identified proteins. The database used was UP000805418 obtained from https://www.uniprot.org/proteomes?query=canis+lupus+familiarisX (taxonomy Canis lupus familiaris). The download date was February of 2023 with a total of 43,622 proteins available. The Perseus analysis software version 2.0.9.0 (Tyanova and Cox, 2018) was used to filter the proteomic results. Matrix proteins identified only by one modification site, as well as those identified by the reverse database and possible contaminants were excluded from subsequent analyses. Proteins were filtered so that only those that had values greater than zero in at least 50% of the samples from at least one of the groups remained in the abundance matrix. Afterwards, a script in the R programming language (https://www.R-project.org/) was used to refine the filter based on the percentage of protein presence in the groups and normalize the data by the total ion count (TIC).
References:
Macron et al., (2020). Data in Brief 31. https://doi.org/10.1016/j.dib.2020.105704
COX, J., MANN, M. (2008) MaxQuant enables high peptide identification rates, individualized p.p.b.-range mass accuracies and proteome-wide protein quantification. Nat Biotechnol; 26:1367–1372.https://doi.org/10.1038/nbt.1511
COX, J., NEUHAUSER, N., MICHALSKI, A., SCHELTEMA, R. A., OLSEN, J. V., MANN, M. (2011) Andromeda: a peptide search engine integrated into the MaxQuant environment. J Proteome Res; 10:1794–1805. https://doi.org/10.1021/pr101065j
TYANOVA, S., COX, J. (2018). Perseus: A Bioinformatics Platform for Integrative Analysis of Proteomics Data in Cancer Research. In: von Stechow, L. (eds) Cancer Systems Biology. Methods in Molecular Biology, 1711. Humana Press, New York, NY. https://doi.org/10.1007/978-1-4939-7493-1_7
Line 174: They wrote that they “rigorously filter the proteomic results (90%)”. If they mean that they required Perseus ANOVA q-values of <=10% between the experimental groups, then this is ok but it needs to be clarified. They need to describe exactly how they did their ANOVAs. Did they treat all seven experimental groups (G1 - G5 with G1 and G2 having paired measurements) as independent experimental conditions? If so, this is ok, but this is not stated, and it needs to be completely and clearly described.
R: Thank you for the observation. Gene Ontology Enrichment Analysis [13] was conduted on the unique features identified within each group, with enriched outputs considered significant at a false discovery rate (FDR) < 0.05, and -log (p value) reported. To assess protein interaction networks, the protein list was submitted to the STRING database [14]. Finally, unique proteins were cross-referenced in the UniProt database. In our figures, we included only proteins with confirmed, predicted, or inferred existence according to their UniProt "Protein Existence" (PE) scores. Proteins categorized as uncertain or Ig-like were excluded to ensure clarity and accuracy. This approach highlights proteins with known or predicted functions, providing a clear and reliable representation of our proteomic data. Statistical calculations were performed for SEVs using GraphPad Prism software (v.10.2.3). All quantitative data were expressed either as the median with interquartile ranges or as means with standard errors, as appropriate. The specific measure used is indicated in each figure legend. For quantitative significance, One-way ANOVA followed by Tukey’s post-hoc test was used for comparison between groups. Comparison between two paired groups was determined by Student's t-test. A p-value of <0.05 was considered statistically significant.
References:
[13] P.D. Thomas, D. Ebert, A. Muruganujan, T. Mushayahama, L.-P. Albou, H. Mi, PANTHER: Making genome-scale phylogenetics accessible to all., Protein Sci. 31 (2022) 8–22. https://doi.org/10.1002/pro.4218.
[14] D. Szklarczyk, R. Kirsch, M. Koutrouli, K. Nastou, F. Mehryary, R. Hachilif, A.L. Gable, T. Fang, N.T. Doncheva, S. Pyysalo, P. Bork, L.J. Jensen, C. von Mering, The STRING database in 2023: protein-protein association networks and functional enrichment analyses for any sequenced genome of interest., Nucleic Acids Res. 51 (2023) D638–D646. https://doi.org/10.1093/nar/gkac1000.
Line 186: The authors wrote “Finally, unique proteins were cross-referenced in the Uniprot database, and squares were inserted in the figures to show only the confirmed, predicted, or inferred proteins while uncertain or Ig-like proteins were not shown.” I can’t understand what they meant. This needs to be rewritten for clarity. Are they referring to the UniProt “PE” (protein existence) scores?
R: Thank you for the observation. The following text was rewritten and is highlighted in the reviewed manuscript.
Finally, exclusive proteins were double-checked in the Uniprot database for Canis lupus familiaris (CANF) and those that retrieved scores as “experimental evidence at protein level”, “experimental evidence at transcript level”, “protein inferred from homology” or “protein predicted” were inserted in squares. Uncertain proteins were excluded as well as Ig-like proteins since they did not retrieve any specific output.
Methods section, “Statistical Analysis” subsection: If this refers to their non-omics data, this needs to be stated here (if not, this type of analysis would be improper for their omics data). They wrote “All quantitative data were expressed either as the median with interquartile ranges or as means with standard errors.” The figure legend(s) need to indicate which they used for each chart.
R: Thank you for the observation. The following text was inserted in the manuscript.
For the non-omics data, values were submitted to Shapiro-Wilk and Kolmogorov-Smirnov normality tests. For the 1st and 2nd collection sets, values were parametric and underwent to paired t-test. For the other sets, Ordinary one-way ANOVA test was performed followed by Tukey’s multiple comparisons test for parametric data or Kruskal-Wallis for non-parametric. Values were expressed as mean and standard deviation (SD).
Line 562: The authors wrote: “Data Availability Statement: Data are available upon request to interested researchers.” Their raw mass spectra data files, their FASTA data file(s) of protein sequences, a ZIP file containing their MaxQuant “/combined/txt/” output files, and any other relevant data all need to be submitted to a public online proteomics data repository such as ProteomeXchange. A table needs to be included clearly mapping the sample-level, LC-MS-level, and MaxQuant files to each other.
R: Thank you for the observation. The data was submitted to the public online proteomics data repository under the following ID: debora.zuccari@famerp.br; passaword limc123.
Additionally, a table (or multiple tables) of their final proteomics data needs to be included with their manuscript as supplemental material or submitted to the online repository with their raw LC-MS data files. Its format should be of an expression table: protein identifiers and gene names (rows), samples (columns), and protein abundance (values), and also the p and q-values from their Perseus ANOVAs.
R: Thank you for the observation. A table with proteomics data was included as supplementary material with the manuscript.
Parts of Figures 3, 4, and 5 are too low resolution, and the labels can’t be read.
R: Thank you for the observation. The resolution of Figures 3,4 and 5 were improved and may be observed in the reviewed manuscript.
Line 182: Gene Ontology Enrichment Analyses require a reference (a.k.a. background) gene set. Typically, it is the entire genome. The authors simply need to state what their reference gene set was.
R: Thank you for the observation. The following text was inserted in the manuscript and is highlighted for better identification.
PANTHER Overrepresentation Test [13] was performed using GO Ontology database (2024-01-17 release) and Canis lupus familiaris gene set as a reference. Only the unique features found in each group were analyzed by Fisher test and outputs having a false discovery rate (FDR) < 0.05 were considered enriched. The -log (p value) was reported.
Reference:   
P.D. Thomas, D. Ebert, A. Muruganujan, T. Mushayahama, L.-P. Albou, H. Mi, PANTHER: Making genome-scale phylogenetics accessible to all., Protein Sci. 31 (2022) 8–22. https://doi.org/10.1002/pro.4218.

Round 2
Reviewer 1 Report
Comments and Suggestions for Authors
I have no further questions.
Author Response
We sincerely thank the reviewers for their thorough attention and thoughtful considerations, which have greatly improved our manuscript.
Reviewer 2 Report
Comments and Suggestions for Authors
Thanks authors for answering the questions and I don't have more comments on the manuscript.
Author Response

(The authors gave the same response as above.)

Reviewer 3 Report
Comments and Suggestions for Authors
The authors completely addressed all of my concerns, and I now recommend that their manuscript be accepted for publication.
Comments on the Quality of English LanguageMinor grammatical errors.
Author Response

(The authors gave the same response as above.)
